# The NF-κB Transcriptional Network Is a High-Dose Vitamin C-Targetable Vulnerability in Breast Cancer

**DOI:** 10.3390/biomedicines11041060

**Published:** 2023-03-30

**Authors:** Ali Mussa, Hafeez Abiola Afolabi, Nazmul Huda Syed, Mustafa Talib, Ahmad Hafiz Murtadha, Khalid Hajissa, Noor Fatmawati Mokhtar, Rohimah Mohamud, Rosline Hassan

**Affiliations:** 1Department of Haematology, School of Medical Sciences, Universiti Sains Malaysia, Kubang Kerian, Kota Bharu 16150, Kelantan, Malaysia; 2Department of Biology, Faculty of Education, Omdurman Islamic University, Omdurman P.O. Box 382, Sudan; 3General Surgery Department, School of Medical Sciences, Universiti Sains Malaysia, Kubang Kerian, Kota Bharu 16150, Kelantan, Malaysia; 4School of Health Sciences, Universiti Sains Malaysia, Kubang Kerian, Kota Bharu 16150, Kelantan, Malaysia; 5Department of Immunology, Doctoral School of Biology, Faculty of Sciences, Eotvos Lorand University, Pázmány Péter sétány 1/C, 1117 Budapest, Hungary; 6Department of Zoology, Faculty of Science and Technology, Omdurman Islamic University, Omdurman P.O. Box 382, Sudan; 7Institute for Research in Molecular Medicine (INFORMM), Universiti Sains Malaysia, Kubang Kerian, Kota Bharu 16150, Kelantan, Malaysia; 8Department of Immunology, School of Medical Sciences, Universiti Sains Malaysia, Kubang Kerian, Kota Bharu 16150, Kelantan, Malaysia

**Keywords:** breast cancer, NF-κB, vitamin C, high dose, chemo- and radioresistance, pro-oxidant

## Abstract

Breast cancer (BC) is the most common cancer type among women with a distinct clinical presentation, but the survival rate remains moderate despite advances in multimodal therapy. Consequently, a deeper understanding of the molecular etiology is required for the development of more effective treatments for BC. The relationship between inflammation and tumorigenesis is well established, and the activation of the pro-inflammatory transcription factor nuclear factor kappa-light-chain-enhancer of activated B cells (NF-κB) is frequently identified in BC. Constitutive NF-κB activation is linked to cell survival, metastasis, proliferation, and hormonal, chemo-, and radiotherapy resistance. Moreover, the crosstalk between NF-κB and other transcription factors is well documented. It is reported that vitamin C plays a key role in preventing and treating a number of pathological conditions, including cancer, when administered at remarkably high doses. Indeed, vitamin C can regulate the activation of NF-κB by inhibiting specific NF-κB-dependent genes and multiple stimuli. In this review, we examine the various NF-κB impacts on BC development. We also provide some insight into how the NF-κB network may be targeted as a potential vulnerability by using natural pro-oxidant therapies such as vitamin C.

## 1. Introduction

Breast cancer (BC) has the highest mortality rate amongst all females around the globe [1]. Indeed, lacking expression of the estrogen receptor (ER), progesterone receptor (PR), and human epidermal growth factor receptor 2 (HER2) makes triple-negative breast cancer (TNBC) the most malignant and metastatic form of the disease [2,3]. Due to genetic heterogeneity of cancer caused by genomic instability, metabolic pathways, molecular genetics, transcriptomics, and deregulated signaling pathways, treating TNBCs is challenging [4,5,6,7,8]. Furthermore, the aggressive TNBC phenotype has been associated with epigenetic modulation, calcium signaling, and the loss of MHC I [9,10,11]. Currently, attenuation of TNBC progression by modulating the cancer signaling pathways such as by inhibiting CDK14, induction of chemokines such as CXCL14, and inhibition of NF-κB is promising [12,13,14]. The NF-κB pathway is highly associated with BC, as shown by its inhibition via parthenolide that inhibits TNBC progression, inducing apoptosis in mammary tumors, enhancing chemo-sensitivity in BC cells, and suppressing metastatic markers in TNBC [15,16,17,18]. The NF-κB pathway also showed potential as a long-term prognostic marker for TNBC via the *SP1* gene and c-erbB2 oncoprotein [19,20]. In addition, NF-κB activation is associated with estrogen-negative breast tumors and, in HER2-positive BC patients treated with paclitaxel, increased NK cells with activated NF-κB are recorded [21,22]. Inhibition of NF-κB and signal transducer and activator of transcription (STAT) with CYT387 successfully interrupts pro-tumorigenic cytokine signaling, thus impairing the TNBC cell line’s growth, indicating the relationship between NF-κB and cytokine signaling [23]. As a predictive therapeutic marker, NF-κB activation via IKKα showed significant association with 10-years relapse-free survival in hormone receptor-positive BC patients [24]. In contrast, in multivariate analysis, NF-κB activation in human breast tumors is correlated with the loss of tumor suppressor protein, 14-3-3σ, which is significantly associated with worse prognosis [25]. Therefore, as shown in a repertoire of studies, targeting NF-κB canonical and non-canonical pathways is crucial in therapeutic cancer management, including BC [26,27]. However, its role in BC development and progression is still under investigation. Here, we examine the complex role of NF-κB impacts on BC development. In addition, we explore how natural pro-oxidant substances, such as vitamin C, may be exploited to target NF-κB networks as a potential vulnerability in BC..

## 2. The Nuclear Factor Kappa-Light-Chain-Enhancer of Activated B Cells (NF-κB)

The NF-κB family of transcriptional regulators is a heterogeneous group that has a central role in regulating a vast range of gene activities that are implicated in various biological processes, such as inflammation, the immunological response, development, apoptosis resistance, and cell growth. These factors are inducible and responsive to cellular signals, allowing them to regulate the expression of genes essential for cellular signaling [28]. The NF-κB members have five related proteins: RelA (p65), RelB, cRel, NF-κB1 (p50), and NF-κB2 (p52). These members participate in targeting the transcription of specific genes by attaching to the κB enhancer component in the DNA, either as homodimers or heterodimers [29]. The family has an essential Rel homology domain (RHD) that enables dimerization, DNA binding, and engagement with IκB antagonists. This domain is named after the reticuloendotheliosis virus (REV) oncogenic protein identified in the T-strain, which is responsible for embryonic lymphatic neoplasms [30]. In addition, RelA, RelB, and c-Rel each feature a transcription activation domain (TAD), but p100 and p105 are devoid of TADs in their structures. These latter proteins serve as precursors to NF-κB, which are subsequently processed to form the p52 and p50 subunits upon degradation [31,32,33]. Normally, suppressive molecules such as IκB, IκBζ, IκBα, IκBβ, IκBγ, IκεB, p105, p100, and Bcl-3, which are components of the ankyrin repeat domain (ARD)-containing proteins, prevent the NF-κB family of proteins from being active in the cytoplasm. These ARD-associated proteins attach to the DNA-binding sequences of the targeted transcription factors, preventing them from being able to initiate the transcription [34,35]. It is critical to emphasize that IκBα is the most essential constituent of the IκB family. This is due to their C-terminal region, which matches the structure of IκB and has suppressive action against NF-κB. The precursor proteins p50, p52, p105, and p100 have IκB-like characteristics [36]. NF-κB is essential for controlling immunological and inflammatory responses. T cell stimulation, maturation, and the effector phase of inflammation are all regulated by NF-κB, which also controls the transcription of numerous pro-inflammatory genes that activate the innate immune cells [37,38] and participates in regulating inflammasome activity [39].

It is not surprising that, in cancer, NF-κB controls cell proliferation, epithelial-to-mesenchymal transition (EMT), survival, invasion, metastasis, and angiogenesis, in addition to the development of cancer stem cells, cellular metabolism, genetic and epigenetic changes, and resistance to therapy in the tumor microenvironment (TME). Additionally, NF-κB activation leads to immunosuppression via several signaling mechanisms.

## 3. The Signaling Pathways of the NF-κB

Two key signaling cascades control the NF-κB stimulation: the canonical pathway and the alternative or non-canonical pathway. The pathways work together to regulate the innate and the adaptive inflammatory responses, in spite of differences in their signaling mechanisms (Figure 1) [34,40].

### 3.1. The Triggering of the Canonical Pathway of NF-κB

The NF-κB canonical pathway can be triggered by several signals that comprise cytokine receptors, pattern recognition receptors, TNF-α receptor family (TNFR) members, and T cell and B cell receptors. It is also triggered by the binding of cytokines such as IL-1β and TNF-α (Figure 1) [41,42].

The main event that drives the induction of the canonical NF-κB is the proteolytic degradation of IκBα, which is started by the phosphorylation of specific amino acid sites. This is brought about by the activation of a multi-subunit IKK complex comprising IKKα and IKKβ catalytic subunits [42] and the IKKγ regulatory subunit, also recognized as an NF-κB essential modulator (NEMO) [43]. IKK induction can be stimulated by a wide set of signals, including chemokines, growth factors, mitogens, microbial components, stress agents, and cytokines [43,44]. Immediately after stimulation, IKK initiates IκBα phosphorylation at two serines (Ser^32^ and Ser^36^) in the N-terminal and thus activates IκBα degradation through a ubiquitin-dependent mechanism facilitated via the proteasome, leading to rapid canonical NF-κB molecules’ nuclear translocation, primarily of the NF-κB1/RelA and NF-κB1/c-Rel dimers [36,45]. These dimers are then transported to the nucleus site and attach to certain DNA promoter or enhancer sequences of the target genes, thereby mediating their activation, including those genes expressing the IκB and the A20 [46,47]. The recently produced IκB in the nucleus subsequently attaches to the NF-κB dimers and disassociates them from their target genes, while the A20 protein remains relatively in the cell cytoplasm and suppresses the TNF-α receptor function [46,47]. In addition, it should be mentioned that the NF-κB pathway contains at least two negative feedback loops, one mediating the IκB cytoplasmic localization and the other associated with A20 protein. Moreover, RelA was found to be highly expressed in ER-negative and ErbB2-positive tumors [48].

### 3.2. The Triggering of the Non-Canonical Pathway of NF-κB

The NF-κB non-canonical pathway is triggered by a limited set of triggers, including the lymphotoxin β receptor (LTβR), the BAFF receptor (BAFFR), receptor activator of nuclear factor-kappa beta (RANK), and CD40, which are all members of the TNFR superfamily. Unlike the canonical pathway, it is not triggered by a wide range of stimuli (Figure 1) [49]. It should be noted that the non-canonical NF-κB stimulation does not require the breakdown of IκBα but rather depends on the degradation of the p52 precursor protein, p100 [34,49]. In addition, a major step in this pathway requires the interaction between IKKα and the NF-κB-inducing kinase (NIK), which together phosphorylate p100, resulting in the ubiquitination and processing of this molecule [50]. Moreover, p100 processing includes the breakdown of its C-terminal IκB-like domain, leading to the production of mature p52 alongside nuclear transfer of the p52/RelB non-canonical NF-κB complex [34,49,51] as heterodimers with RelA or RelB and this translocation is facilitated via nuclear localization signals (NLSs) that mediate the binding of RelB to importin alpha5 and alpha6, followed by subsequent attachment of p52/RelB on the promoter sequence of specific NF-κB target genes to induce their transcriptional activity [51,52,53,54]. The regulation of non-canonical genes varies depending on the cell type, such as *SCF/BLC/ELC* lymphokines in dendritic cells and *Naf-1* in epithelial cells. It is important to note that both the canonical and non-canonical NF-κB cascades have a significant influence on the development and advancement of cancers, including BC [55]. Recent reports have concluded that ERα has the ability to down-regulate the activity of RelB in BC cells. The precise machinery by which ERα regulates RelB has not been fully investigated, but it is thought to involve direct interaction between the two proteins. One study suggested that ERα binds to the *RelB* promoter, leading to suppression of RelB expression and its downstream target Bcl-2 [56]. Another study found that ERα can directly interact with the phosphorylated form of RelB and inhibit its nucleus translocation, thereby blocking the activation of its target genes [57]. This suggests that the canonical NF-κB induction could be elevated in ER-negative patients.

## 4. NF-κB in BC

When genes or signaling pathways are mutated or altered in BC, NF-κB may become permanently activated, promoting cell growth and survival despite DNA damage and other stresses. This can lead to the formation of tumors and the spread of cancer cells throughout the body. Multiple pathways have been identified that may activate NF-κB in BC cells. For example, mutations in certain genes, such as *BRCA1* and *BRCA2*, can significantly result in induction of the NF-κB [58]. Additionally, certain signaling pathways, such as the phosphoinositide 3-kinase (PI3K)/Akt/mitogen-activated protein kinase (MAPK) cascades interact with NF-κB and can activate each other reciprocally in BC cells.

### 4.1. NF-κB Activation in BC

A recent report showed that NF-κB can activate a signaling pathway known as PI3K/Akt/extracellular signal-related kinase (ERK1/2), which in turn can lead to the production of an oncoprotein called mucin (MUC-1 and MUC1-C). This oncoprotein has the capability to interact with other receptors such as HER2 and epidermal growth factor receptor (EGFR) which are tyrosine kinase receptors. These interactions can contribute to the growth and survival of BC cells. However, blocking the activity of this signaling system has been shown to successfully decrease the proliferation of BC cells, indicating that targeting this route may be a feasible treatment method for BC [59]. EGFR in turn can stimulate and induce a cascade of signaling networks, such as RAS/RAF/MEK/ERK and ERK/Akt/NF-κB, which are critical signaling pathways in the MAPK cell signaling networks. These pathways play a significant role in the growth of BC [60,61,62,63]. ERK activation can further activate and phosphorylate a number of signaling pathways that are under its control, such as NF-κB, c-MYC, STATs, SAP-1a, AP-1, Elk-1, cyclin D1, a cyclooxygenase (COX-2), and the ER. As a consequence, multiple genes implicated in cell growth, survival, and proliferation—all characteristics of tumors—are under the transcriptional control of this process [63,64,65,66]. The membrane-bound ER (mbER) complex, which consists of SRC and PI3K, interacts with NF-κB, growth factor signaling, and the ER [67]. When the mbER is bound to estrogen, it increases EGFR tyrosine kinase activity and activates mTOR through PI3K-mediated Akt phosphorylation. Furthermore, phosphorylated Akt phosphorylates both ligand-dependent and independent ERs. Additionally, when the ER binds to estrogen, it activates other pathways including MAPK and PI3K/Akt/mTOR to enhance cell survival, proliferation, and angiogenesis. In MCF-7 cells expressing ERs, HER2 may interact with the NF-κB pathway [67]. In response to ionizing radiation, HER2 overexpression increases the activity of the PI3K/Akt pathway, which activates NF-κB and up-regulates pro-survival genes including manganese superoxide dismutase (MnSOD), cyclin B1, and HER2 [67]. Moreover, HER2 can mediate the stimulation of the canonical NF-κB, then subsequently mediates the formation and maintenance of the EMT [68]. The initiation of EMT involves the induction of numerous transcription regulators, including the Smad-interacting proteins ZEP-1/ZFHX1A, ZEB-2/ZFHX1B, SLUG, SIP1, and TWIST1 by the RelA molecule [68]. Likewise, estrogen increases the transcription of *IKK-α*, *ERs*, and *steroid coactivator 3 (SRC3)* by binding to a particular area on the *cyclin D1* promoter. IKK-α subsequently stimulates the phosphorylation of ERs and SRC3 through a positive feedback loop. It has also been observed that IKK-α interacts with and activates E2F1 and p300/CBP-associated factor (PCAF), and acetylates amino acids on histone H3, resulting in sustained BC cell cycle activity [69]. Remarkably, NF-κB has a DNA binding site in the promoter sequence of *matrix metalloproteinase-9 (MMP-9)* gene. When NF-κB attaches and interacts with this promoter sequence, *MMP-9* gene transcription is actively induced [70]. MMP-9, in turn, can strongly promote EMT induction, and also activate transforming growth factor-β (TGF-β) as well as EGFR activation [71], resulting in BC growth, metastasis, development, and drug resistance. Treatment with NF-κB inhibitors effectively reversed this action. Indeed, NF-κB can assemble, interact with, and activate various anti-apoptotic proteins which mediate BC resistance to apoptosis. For example, through a TNF-α/ER/NF-κB-dependent mechanism, TNF-α activates and up-regulates various anti-apoptotic proteins, such as baculoviral IAP repeat-containing 3 (BIRC-3) and cellular inhibitor of apoptosis protein (cIAP-2). Furthermore, NF-κB can activate genes that confer autophagy resistance [69]. The X-box binding protein-1 (XPB-1) transcription factor activates the unfolded protein response (UPR), which is required for endoplasmic reticulum stress resistance. Upon activation with NF-κB, a spliced form called XPB-1 (S) is generated that successfully mediates estrogen-dependent and estrogen-independent resistance of BC; in addition, it also induces the up-regulation of various anti-apoptotic proteins including B cell lymphoma-2 (Bcl-2), HER2, Bcl-xl, Ras, c-Fos, c-Jun, and mutated p53 [69,72], and this suggests that NF-κB contributed to BC survival and resistance.

Conversely, NF-κB interacts with and activates a number of proteins that are beneficial to BC, including interleukin 6 (IL-6), hypoxia-inducible factor 1 alpha (HIF-1α), STAT3, and vascular endothelial growth factor (VEGF). Interactions between these proteins have been found to contribute to the BC progression in various ways: NF-κB and IL-6 are known to have a positive feedback loop, where NF-κB activates the transcription of genes that encode for *IL-6*, which in turn activates NF-κB [73]. This results in a sustained pro-inflammatory response and promotes cancer cell growth and survival. HIF-1α is known to induce the *IL-6* gene, which can then activate the NF-κB [74]. Moreover, HIF-1α can also activate NF-κB directly. STAT3 is in turn activated by IL-6 and NF-κB [75], and it can also activate the *HIF-1α* gene. STAT3 can also constrain the activity of NF-κB by repressing the induction of NF-κB target genes [74]. Finally, HIF-1α and STAT3 can also cooperate to drive cancer cell survival and growth through induction of VEGF, interleukin 6 receptor (IL-6R), HER2, c-Src, PI3K, and JAK/STAT [74,76]. All these interactions between these proteins ultimately result in the enhancement of the BC’s overall development. Inhibiting the activity of these proteins is a promising approach for the therapy of BC. Drugs that target these proteins have been established and are in clinical testing. Interestingly, BC cells may stimulate their own growth and development by secreting various cytokines, including colony-stimulating factor-2 (CSF2), monocyte chemoattractant protein-1 (MCP-1), IL-6, and IL-8. Through an autocrine positive feedback loop mechanism, these cytokines boost the induction of the NF-κB and Wnt/β-catenin, which together enhance the production of additional cytokines, resulting in BC treatment resistance and promoting the formation of cancer stem cells [77].

Additionally, NF-κB may help BC cells evade the immune system by inducing a number of proteins, such as programmed cell death receptor 1 ligand (PD-L1). In fact, NF-κB overexpresses the *PD-L1* gene at the transcriptional level via direct interaction with p65 that binds to a specific sequence on the *PD-L1* gene promoter [78]. Cytokines of various types (TNF-α, interferon gamma (IFN-γ), and IL-17) and their interaction with NF-κB are reported to overexpress *PD-L1* [78]. Additionally, *PD-L1* transcription is induced via a process requiring MUC-1-mediated NF-κB activation [68]. Moreover, EGFR was also reported to induce *PD-L1* transcription through its interaction with NF-κB. In fact, EGFR phosphorylates IκBα and activates ERK and Akt, which together up-regulate *HIF-1α* and *PD-L1* [78]. Mechanistically, the *PD-L1* gene promoter also contains a binding site for c-MYC. Following activation of c-MYC by NF-κB, the activated transcription factor effectively promotes *PD-L1* overexpression [78]. Altogether, it seems that NF-κB and its interaction with several signaling pathways are important for BC immunological resistance. However, in order to develop an effective treatment plan, it is necessary to fully understand the significant role of the NF-κB network in the pathophysiology of BC. Figure 2 shows how the NF-κB networks are activated and how they interact intricately with different signaling pathways to support BC growth and development.

### 4.2. NF-κB Activates HIF-1α in BC

When the oxygen (O_2_) levels in the tumor microenvironment fall below 5–10 mmHg, it is referred to as hypoxia, which is characterized by discrepancies in O_2_ absorption and consumption induced by the uncontrolled proliferation of cancer tissues [79]. It should be noted that hypoxia is a characteristic of many solid tumors, including BC, and that it is connected to cell survival, proliferation, angiogenesis, and metastasis [79]. Importantly, hypoxia promotes the activities of clinically significant transcription factors (HIFs), which are represented by: (1) a heterodimer structure composed of O_2_-senstive α subunit-dependent transcription factors―HIF-1α, HIF-2α, and HIF-3α―containing an O_2_-dependent degradation domain (ODDD) that mediates the proteasomal destruction of the HIFs through the inhibition of the N-terminal domain (N-TAD) and (2) HIF-1β transcription factors that do not require an α subunit [79,80]. Significantly, after HIF-1α overexpression, it binds to p300/CBP and activates various genes in favor of BC by binding to a specific gene region called the hypoxia-responsive element (*HRE*). However, during normoxic conidiations, HIFs are tightly controlled and cannot perform their function. In this process, two enzymes belonging to the oxo-glutarate-dependent di-oxygenase family, prolyl hydroxylase-2 (PHD-2) and factor-inhibiting HIF-1α (FIH), utilize O_2_, ferrous ions (Fe^2+^), 2-oxo-glutarate, and ascorbic acid as enzyme cofactors for their optimal activity to mediate HIF-1α regulation by hydroxylation of Pro^402^ and Pro^562^. These two enzymes hydroxylate multiple amino acid residues on HIF-1α and promote its proteasomal degradation through a mechanism involving the von Hippel–Lindau protein (pVHL)-mediated E3 ubiquitin ligase [80]. Here, it should be noted that the optimal functions of these two enzymes mainly depend on normoxic conditions. However, during hypoxia the two enzymes are inactive, thus HIF-1α mediates the transcription of thousands of genes in BC such as *MMP-9*, *VEGF* [81], *glucose transporter 1 (GLUT-1)*, *TGF-β*, *CXCR4*/its ligand *CXCL12*, and *Bcl-2* [82]. Conversely, various reports support the fact that there is a clear connection between NF-κB and HIF-1α activation. This is supported by the fact that NF-κB has a binding site (−97/−188 bp) at the *HIF-1α* promoter sequence region. Indeed, the p50 and p65 of the NF-κB subunits have been found to directly bind to the −197/−188 bp region, activating the promoter and inducing *HIF-1α* transcription. Moreover, TNF-α-mediated NF-κB activation was found to stabilize HIF-1α protein levels [83]. Similarly, HIF-1α can activate NF-κB by phosphorylating p65 at Ser^276^ and IκB hyper-phosphorylation, increasing its nuclear accumulation, resulting in the activation of various target genes [83]. In addition, NF-κB and HIF-1α appear to activate each other indirectly and reciprocally. For instance, NF-κB activation stimulates PI3K/Akt/ERK1/2-MAPK, which in turn promotes the activation of EGFR and HIF-1α. EGFR also has the ability to activate HIF-1α and RAS/RAF/MEK/ERK as well as ERK/Akt/NF-κB and PI3K/Akt/mTOR which subsequently activate more HIF-1α [60,61,62,63,78]. Remarkably, both NF-κB and HIF-1α are tightly controlled in the same manner due to their shared pathways, including IKKβ being a target of PHDs. This shows that hypoxia may enhance NF-κB stimulation in response to hypoxia-induced PHD suppression. Hypoxia may also reduce the NF-κB regulatory function by decreasing PHD-mediated IKKβ hydroxylation. In addition, IκBα and p105 have been discovered to be ARD-containing proteins that are hydroxylated at particular sites by FIH, but the relevance of this process is still elusive [83]. Ultimately, it seems that the NF-κB and HIF-1α communication is essential for the development of solid cancers, particularly BC. Due to the complexity of the connection between NF-κB and HIF-1α, it is difficult to appreciate and deconstruct this crosstalk, which necessitate the exploration of various signaling pathways and the possible positive and negative feedback loops for effective targeting and therapy in BC. Figure 3 shows how HIF-1α is activated and how it interacts with NF-κB, then how they subsequently activate each other, thus supporting BC development. Yet, the picture also demonstrates how reactive oxygen species (ROS) are employed by NF-κB and HIF-1α to promote BC development and how high doses of vitamin C may be used to efficiently target and block this intricate network.

## 5. NF-κB Recruits and Activates Immunosuppressive Cells in BC Tumor Microenvironment

### 5.1. Tregs

Regulatory T cells (Tregs), thymic regulatory T cells (tTregs), and peripheral regulatory T cells (pTregs) maintain immune homeostasis by inducing self-tolerance and arresting the immune response. Treg cells have been phenotyped as interleukin receptor-2α (CD25)-expressing and forkhead box P3 (FoxP3)-expressing CD4^+^ T cells [84,85].

Indeed, Treg cells are incorporated into tumor sites by cancer type-specific chemokines induced by tumor cells, in addition to cells such as myeloid-derived suppressor cells (MDSCs) and tumor-associated macrophages (TAMs) [86]. For instance, CCL20, CCL17, and CCL22 have been described to recruit Treg cells in the TEMs of non-small cell lung cancer (NSCLC), colorectal cancer (CRC), and muscle-invasive bladder cancer (MIBC) [87]. However, in BC, ligands associated with the receptors (CCR5, CCR8, CCR10, CX3CR1, CXCR3, and CXCR6) have been described for Treg cell recruitment [88]. Therefore, high Treg cell numbers were associated with the progression of the disease in cancer patients, particularly those with gastric cancer, ovarian cancer, or BC, despite the fact that a favorable regression was induced for CRC, head and neck squamous cell carcinoma (HNSCC), and Hodgkin’s lymphoma (HL) [89]. This might be attributable to the fact that the TME complexity differs between cancer subtypes, as was revealed in BC subtypes and that infiltrating Tregs were linked with poorer disease progression in ER-positive BC tumors [90].

By secreting large quantities of immunosuppressive cytokines such as TGF-β, IL-10, IL-35, granzyme B, and perforin or by activating inhibitory immune checkpoints such as cytotoxic T lymphocyte-associated antigen-4 (CTLA-4), programmed cell death protein-1 (PD-1), lymphocyte activation gene-3 (LAG-3), and PD-L1, Treg cells are constitutively activated and are highly immunosuppressive within the TME, down-regulating natural killer (NK) cells, effector CD4^+^ or CD8^+^ T cells, and activated B cells [88,89]. Therefore, Tregs provide a favorable setting for tumor growth. Accordingly, a wide range of molecular mechanisms have been proposed as contributing to Tregs’ function in BC. As such, NF-κB was studied due to its involvement in controlling Tregs’ development and function [91]. Indeed, NF-κB was described to regulate de novo expression of FoxP3 in developing thymocytes, maintaining the maturity and identity of Tregs [92,93].

The oncogenic effect of NF-κB in BC was associated with cell proliferation induction by activating granulocyte–macrophage colony-stimulating factor (GM-CSF), IL-2, cyclin D1, and CD40 ligand (CD40L). In addition to promoting metastasis and angiogenesis, NF-κB acts to suppress apoptosis by turning on genes that have an anti-apoptotic effect, for instance, members of Bcl-2 (including A1/BFL1 and Bcl-x), IL-1β-converting enzyme (FLICE) inhibitory protein (c-FLIP), caspase-8/FAS-associated death domain (FADD), and c-IAPs [91,94,95].

### 5.2. MDSCs

MDSCs are strongly associated with BC pathogenesis, therefore, the depletion of MDSCs was proposed as an effective pathway for treating mammary carcinoma [96]. The activation of NF-κB within MDSCs contributes to down-regulation of the effective immune response within the TME. Several studies showed that TLRs induce NF-κB activation through MyD88 and the exosomal HSP70 pathway, and the up-regulation of IL-1β on tumor cells increases the number of peroxynitrite-producing MDSCs via the NF-κB pathway. In addition, the interaction of TNF-α with its ligand increases iNOS expression in a way that is reliant on both NF-κB and p38 MAPK, which ultimately leads to an increase in the suppressive activity of MDSCs [97]. Collectively, these results pointed to the substantial role that NF-κB pathways play in the aforementioned kinds of cells, suggesting that these pathways might be viable therapeutic targets.

### 5.3. TAMs

Tumor-associated macrophages (TAMs) are associated with tumor induction, regardless of their tumoricidal effect. Originally, these cells are derived from circulating macrophage precursors. These precursor cells within the TME can be differentiated into active macrophages (M1 type) when stimulated by a classical pathway such as lipopolysaccharide (LPS) and IFN-γ, which exerts an antitumor effect by releasing IL-1β and cytolysin A (ClyA) or type 2 macrophages (M2) that are generated by the induction of IL-10 and TFG-β and positively enhance tumorigenesis. Moreover, the gathered findings demonstrated that TAMs found in tumor settings had a tendency to develop into M2 types when they interacted with the TME [98].

The transition from the M1 to the M2 subtype reduces the capacity of macrophages for natural identification and phagocytosis, in addition to the effectiveness of CD4^+^ and CD8^+^ T lymphocytes in the elimination of tumors. Similarly, it may stimulate Treg cells, which can lead to immunosuppression of the tumor. This is primarily caused by the expression of PD-L1, which initiates a crosslink with PD-1 in T cells, thereby reducing the tumoricidal effect. This can lead to the stimulation of T cells that suppress the immune defense. In addition, TAMs may generate suppressive abilities by generating TNF-α via the JAK/STAT3 and PI3K/Akt signaling pathways. This results in an increase in the expression of PD-L1. Likewise, it was mentioned that IL-6 was connected to the development of TNBC by promoting the expression of *PD-L1* [78].

Moreover, tumor escape is further facilitated by TGF-β, which induces M2 polarization and up-regulates *PD-L1* expression inside the TME, hence increasing TAM inhibitory action. In addition, TAMs may foster the expansion and function of PD-1^+^Tregs, which aid in tumor immune evasion in TNBC [98]. Since the *PD-L1* promoter contains numerous NF-κB-interacting motifs, it has been hypothesized that activation of this transcription factor is necessary for *PD-L1* expression through the aforementioned pathways [78]. Figure 4 describes the significant impact of NF-κB expression that leads to the expression of various BC-related proteins and activates various immunosuppressive cells.

## 6. High-Dose Vitamin C Targeting the NF-κB Transcriptional Network

Antioxidants are substances that may protect cells against ROS and may play a role in the progression of certain conditions, such as cancer. Antioxidants interrupt the process of “electron stealing” by donating one of their own electrons to neutralize free radicals [99], converting them into an inert product. However, during normal physiological conditions, to prevent damage, cells counteract the oxidative stress that is generated by free radicals by synthesizing enzyme proteins such as superoxide dismutase (SOD) and catalase (CAT) or non-enzymatic compounds such as *N*-acetyl-cysteine (NAC) and glutathione (GSH), to block the effect of the oxidative reactions’ chains [99]. Antioxidant therapies such as vitamin E, vitamin C, selenium, and β-carotene have been used widely in cancer patients to minimize the side effects of chemotherapeutic drugs that generate high concentrations of ROS [99,100]. On the other hand, ROS plays a central role in maintaining the balance of a cell’s redox state. The levels of ROS—O_2_^•−^, HO^•^, NO^•^, N_2_O^•^, LOO^•^, H_2_O_2_—are defined by the balance between their generation and elimination by antioxidants. The redox state, or overall reduction potential, of a cell is influenced by the concentration of reducing species such as GSH, NADH, and FADH^2^ [99]. Under normal physiological conditions, the redox state is maintained at increasingly negative values. The redox state is a quantitative indicator of the capacity to decrease all redox couples, including GSSG/GSH and NAD^+^/NADPH, identified in a given biological fluid, organelle, cell, or tissue [99]. However, when ROS levels surpass the capacity of antioxidants to eliminate them, this leads to “oxidative stress,” which can cause cellular damage (DNA, proteins, and lipids) if they are high and persistent. Although past studies have shown that low concentrations of ROS contribute to the development of cancer, new studies indicate that this is not the case [101]. In this regard, high doses of vitamin C have been proposed for cancer therapy, utilizing its pro-oxidant properties to cause significant oxidative stress through the production of large amounts of ROS rather than its antioxidant properties [102,103]. In fact, vitamin C has been widely applied to treat various cancers (solid and hematological), in vitro, in vivo, and in patients [103]. The vitamin has been shown to be very effective, despite some debate regarding the findings and the need for more research.

### The Pro-Oxidant Activity of Vitamin C

Vitamin C has two forms—de-hydroascorbic acid (DHA) and ascorbic intermediate radical (Asc^•−^)—outside and inside cancer cells after its oxidation by ROS, depending on the receptor that it encounters, GLUT-1 or sodium-dependent vitamin C transporters (SVCTs) [104]. Once vitamin C enters the cell it interacts with cellular transition metal Fe^2+^ ions (labile iron) to produce ferric ions (Fe^3+^) through a mechanism involving the Fenton reaction to produce H_2_O_2_ and HO^−^, then via the Haber–Weiss reaction it produces more O_2_^•−^, H_2_O_2_, and HO^•^ (Figure 3) [104]. These interactions produce massive concentrations of ROS which eventually result in cancer cell death.

Indeed, it has been shown that ROS interacts with NF-κB and induces its nuclear translocation via a variety of mechanisms (Figure 3). Study results show that low concentration of H_2_O_2_ effectively activate the p50/p65 heterodimer in the cytoplasm. Subsequently, the research uncovered that H_2_O_2_ induces NF-κB only when it decomposes into O_2_^•−^ and HO^•^. This induction was blocked by NAC treatment [105]. Conversely, another study showed that, in response to oxidative stress after addition of phenobarbital, NF-κB activation increased significantly and H_2_O_2_ was thought to be an important factor in this process, while the addition of vitamin E blocked this activation [106]. Interestingly, upcoming reports have shown that TNF-α induced ROS [107], which mediated the phosphorylation of RelA at Ser^276^ through a mechanism-dependent protein kinase (PKAc). Using different antioxidants including vitamin C effectively inhibited the activation of NF-κB [107,108]. RelA’s phosphorylation is crucial and necessary for the protein to bind to its target genes [109]. These results show that NF-κB responds positively to oxidative stress; for this reason, it is also known as the “oxidative stress-responsive transcription factor.” However, it appears that NF-κB uses this redox imbalance state and other factors to interact with HER2, inducing pro-survival proteins and promoting the translation of various proteins such as MnSOD, cyclin B1, and HER2. MnSOD can generate H_2_O_2_ from O_2_^•−^, which prevents oxidative damage, modifies apoptosis, and ultimately causes radiotherapy drug resistance [110]. Furthermore, ROS were discovered to phosphorylate p65 and activate p50, resulting in *PD-L1* overexpression (Figure 3) [111]. Another study found that epidermal growth factor (EGF) treatment significantly increased ROS generation, which induced the P13K/Akt/NF-κB signaling cascade and promoted EMT. However, metformin treatment inhibited this cascade (Figure 3) [112]. Conversely, H_2_O_2_ generated from external sources or by the enzyme NOX4, which is activated by thrombin, causes an increase in the levels of ROS, which subsequently transcriptionally promote *HIF-1α* via a unique binding site in the promoter, via the NF-κB pathway (Figure 3) [113]. This activation by ROS was independent of oxygen levels, which enables the early initiation of BC (early stages). In addition, by inhibiting the enzyme SOD1 with diethyldithiocarbamate (DDC), the intracellular levels of O_2_^•−^ are increased, which in turn enhances the p50 and p65 nuclear localization. This leads to the transcriptional activation of Bcl-xl and X-linked inhibitor of apoptosis (x-IAP), as well as promotion of the process of EMT. The addition of JSH-23 successfully inhibited the localization of p65. Furthermore, the increased levels of O_2_^•−^ and the involvement of peroxynitrite (ONOO^−^) together phosphorylate IKKβ and IκBα at two serines, 32 and 36, which results in complete degradation of IκBα. This suggests that this sustained IKKβ phosphorylation is due to a redox imbalance. Additionally, O_2_^•−^-mediated NF-κB activation enhances MMP activity, which leads to the migration, invasion, and survival of BC cells [114]. However, some data point to the possibility that the ROS-mediated oxidative stress employed by NF-κB in cancer growth might be used to inhibit NF-κB’s oncogenic activities. A practical strategy would be the powerful pro-oxidant activity of high-dose vitamin C, which produces a significant quantity of ROS that leads to an increased redox imbalance [103]. It has been shown that, via TNF-α and IL-1β, vitamin C blocks the phosphorylation of the IKK protein [115]. In line with this, the oxidized form of vitamin C, DHA, was found to restrict the phosphorylation of both IκBα and IKK [116]. Recently, vitamin C alone or in combination with other therapies has been utilized to target NF-κB activity in a variety of in vitro and in vivo cancers, such as skin cutaneous melanoma (SKCM), lung adenocarcinoma (LUAD), esophageal cancer, cervical cancer, CRC, melanoma, and thyroid carcinoma (THCA) [104]. For instance, when esophageal cancer cells were treated with 5-fluorouracil (5-FU) and cisplatin, the nuclear localization of p65 increased. Prior therapy with high-dose vitamin C, on the other hand, was able to attenuate this effect [104], suggesting that 5-FU could influence ROS in favor of NF-κB [117]. Surprisingly, vitamin C was reported to decrease NF-κB function and increase *p53* overexpression and stability [104]. However, the precise mechanism by which vitamin C activates p53 in order to suppress NF-κB is elusive and requires more investigation. Vitamin C reduced TNF-α-mediated induction of the NF-κB in BC. This was backed by the evidence that DHA prevented p50 nuclear translocation and IκBα phosphorylation; more crucially, DHA inhibited NIK and IKKβ activation independent of p38-MAP kinase [118]. Supporting this notion, accumulation of intracellular DHA interacted directly with IKKβ, thus rendering the protein inactive as well as decreasing the activity of NIK protein (Figure 3) [119]. Likewise, pyridylporphyrin (MnP) and vitamin C together were found to be effective in BC inhibition. MnP causes vitamin C oxidation, resulting in significant H_2_O_2_ production. In response, H_2_O_2_ inhibited the phosphorylation of both NF-κB and ERK1/2 independent of p38-MAP kinase, leading to a drop in x-IAP associated with PARP cleavage and resulting in caspase-dependent apoptosis. H_2_O_2_ activation of AIF (a mitochondrial flavoprotein) caused caspase-independent apoptosis. Although the results are encouraging, the researchers did not identify which NF-κB subunits were suppressed [120]. However, there is a discrepancy in which component of ROS may effectively inhibit NF-κB, and the bulk of the data have solely focused on the canonical route. Studies that investigate both canonical and non-canonical NF-κB in depth and identify the specific part of ROS that might be the most suppressive component in BC are urgently required.

As previously stated, vitamin C is an essential cofactor for the efficient functioning of PHD2 and FIHs (Figure 3). During hypoxia, vitamin C interacts with O_2_, Fe^2+^, and 2-oxoglutarate to restore the activity of PHD2 and FIH, therefore inhibiting HIF-1α activity. As HIF-1α is a crucial transcription factor that drives *NF-κB* overexpression, vitamin C will eventually result in NF-κB suppression by inhibiting HIF-1α (Figure 3). Conversely, this could probably be the case for PD-L1 too, since NF-κB and HIF-1α are significant transcription factors for PD-L1 expression. Indeed, these data lead to the hypothesis that vitamin C-deficient cells can have increased *NF-κB*, *HIF-1α*, and *PD-L1* expression, potentially enhancing BC development.

Remarkably, NF-κB may protect BC cells from ROS produced by vitamin C and stimulate genes that mediate crosstalk between NF-κB and JNK. This intercellular communication inhibits persistent JNK activity [121]. Another mechanism by which NF-κB may protect cells from ROS is by promoting the production of various antioxidative enzymes, including dihydrodiol dehydrogenase (DDH1), HO-1, CAT, manganese superoxide dismutase (MnSOD), glutathione peroxidase-1 (Gpx1), copper–zinc superoxide dismutase (CU-Zn-SOD), ferritin heavy chain (FTH), thioredoxin-1, 2 (Trx1 and Trx2), glutathione S-transferase Pi (GST-pi), NADPH-dehydrogenase-quinone-1 (NQO1), and metallothionein-3 (MT3) (Figure 3) [121]. The above data indicate that NF-κB can negatively regulate the amount of ROS that are generated via the pro-oxidant activity of vitamin C, thus mediating ROS resistance by BC cells.

## 7. Conclusions and Feature Perspectives

The transcription factor NF-κB is an essential component in the stimulation of a wide variety of transcriptional networks that lead to the development and progression of multiple cancers, including BC. In BC, aberrant NF-κB activation is an indication of increased hormonal therapy resistance, chemotherapy resistance, and radiotherapy resistance. Thus, blocking NF-κB in BC with high-dose vitamin C seems to be an effective therapeutic strategy. As such, upcoming studies should place emphasis on whether inhibiting NF-κB with high-dose vitamin C could sensitize BC cells to hormonal therapy, chemotherapy, and radiation treatment. Moreover, the bulk of existing NF-κB inhibitors seem to behave more as chemotherapy sensitizers than as anti-cancer medicines [26]; this is likely due to the fact that the transcriptional network associated with NF-κB activation is very complicated and does not rely on a single pathway for activation. In turn, this could restrict the use of a single pharmacological agent to suppress NF-κB. As a result, using drugs that target the key pathways that activate NF-κB in BC might be a viable option. Moreover, several signaling pathways, including PI3K/Akt/ERK, RAS/RAF/MEK/ERK, PI3K/Akt/mTOR, and JAK/STAT/IL-6, play critical roles in NF-κB activation, which leads to global drug resistance, invasion, metastasis, and progression of BC. Hence, a combination of high-dose vitamin C, an anti-inflammatory medication, proteasome inhibitors, and inhibitors of PI3K/Akt/ERK, RAS/RAF/MEK/ERK, and PI3K/Akt/mTOR might be an effective therapy approach for inhibiting NF-κB. As consequence, this may extend NF-κB inhibition, diminish the impact of inflammatory TME, and therefore sensitize BC cells to radiation therapy and chemotherapy, ultimately inhibiting cancer growth and progression.

Conversely, low concentrations of ROS may cause oxidative stress, which in turn activates NF-κB, and subsequently leads to the translation of proteins that promote the survival, migration, and invasion of BC cells. However, the same oxidative stress can also be harnessed as a tool to suppress NF-κB activity in favor of cancer treatment. Furthermore, the double role of ROS in NF-κB activation demonstrates the intricate landscape of oxidative stress in cancer development and emphasizes the importance of comprehending its mechanisms in the fight against BC. This is because oxidative stress plays a vital role in both the development and progression of cancer. By using this information, it may be possible in the future to design targeting medicines in addition to the existing pro-oxidant medications such as vitamin C that take advantage of the interaction involving oxidative stress and NF-κB to inhibit the growth and metastasis of BC cells in vivo as well as in vitro and later in clinical trials. Indeed, modest levels of ROS often trigger the activation of NF-κB, HIF-1α, and PD-L1. Studies are urgently needed to determine if HIF-1α and PD-L1 may be efficiently inhibited in response to NF-κB inactivation or directly by high-dose vitamin C-mediated high concentrations of ROS. In fact, NF-κB stimulates many target genes (e.g., *SODs*, *FTH*, *Trxs*, *GST-pi*, *NQO1*, and *MT3*), which might possibly suppress ROS production by high-dose vitamin C, lowering its anti-cancer activity significantly. It would be fascinating to see whether inhibiting these genes may enhance the anti-NF-κB effects of high-dose vitamin C. Finally, vitamin C may block cytokine-triggered signaling pathways, particularly those that have TNF-α as a common activator and may be stimulated by NF-κB to promote BC, and activate Tregs, TAMs, and MDSCs. However, more studies are required to incorporate these notions.

## Figures and Tables

**Figure 1 biomedicines-11-01060-f001:**
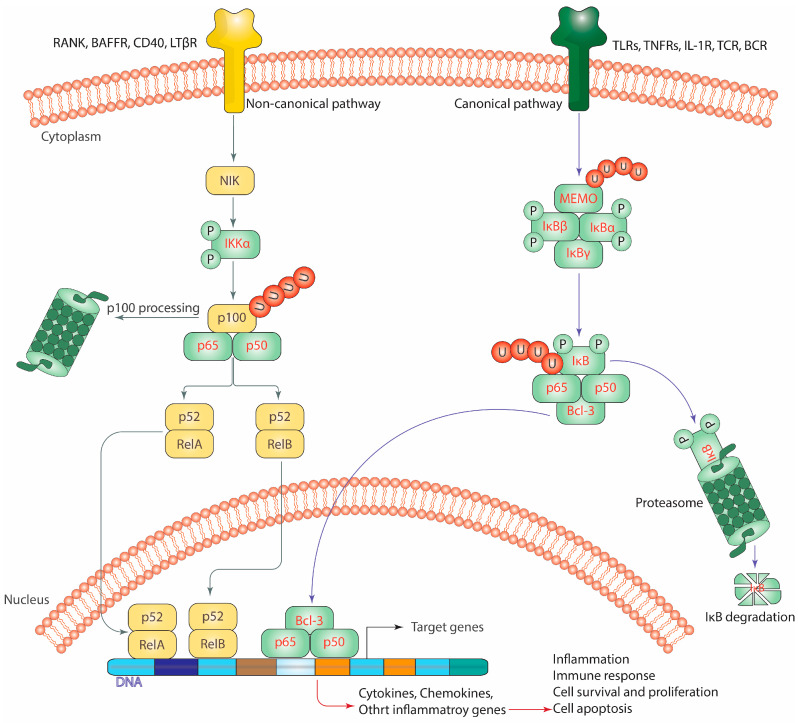
The canonical and non-canonical pathways of the NF-κB. The regular activation of NF-κB. TLRs, Toll-like receptors; TNFRs, TNF-α receptors; TCR, T cell receptor; BCR, B cell receptor; LTβR, lymphotoxin β receptor; BAFFR, the BAFF receptor; RANK, receptor activator of nuclear factor-kappa beta; MEMO, NF-κB essential modulator; NIK, NF-κB-inducing kinase. Note: For more information, please see the text.

**Figure 2 biomedicines-11-01060-f002:**
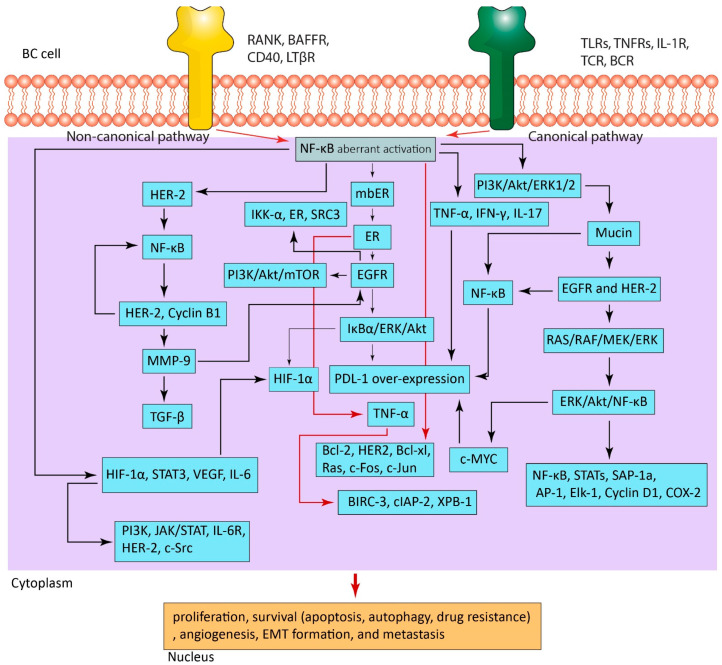
The activation of the NF-κB in breast cancer microenvironment. NF-κB interacts with and activates various signaling pathways, which eventually result in BC progression. TLRs, Toll-like receptors; TNFRs, TNF-α receptors; TCR, T cell receptor; BCR, B cell receptor; LTβR, lymphotoxin β receptor; BAFFR, the BAFF receptor; RANK, receptor activator of nuclear factor-kappa beta; BC cell, breast cancer cell; PI3K, phosphoinositide 3-kinase; ERK, extracellular signal-related kinase; HER2, human epidermal growth factor receptor 2; ER, estrogen receptor; NF-κB, nuclear factor-kappa beta of activated B cell; EGFR, epidermal growth factor receptor; mbER, membrane-bound ER; SRC3, steroid coactivator 3; MMP-9, matrix metalloproteinase-9; TGF-β, transforming growth factor-β; BIRC-3, baculoviral IAP repeat-containing 3; cIAP-2, cellular inhibitor of apoptosis protein; XPB-1, X-box binding protein-1; Bcl-2, B cell lymphoma-2; IL-6, interleukin 6; IL-17, interleukin 17; HIF-1α, hypoxia-inducible factor 1 alpha; VEGF, vascular endothelial growth factor; IL-6R, interleukin 6 receptor; EMT; epithelial-to-mesenchymal transition; PD-L1, programmed cell death receptor 1 ligand; IFN-γ, interferon gamma; STAT, signal transducer and activator of transcription. Note: For more information, please see the text.

**Figure 3 biomedicines-11-01060-f003:**
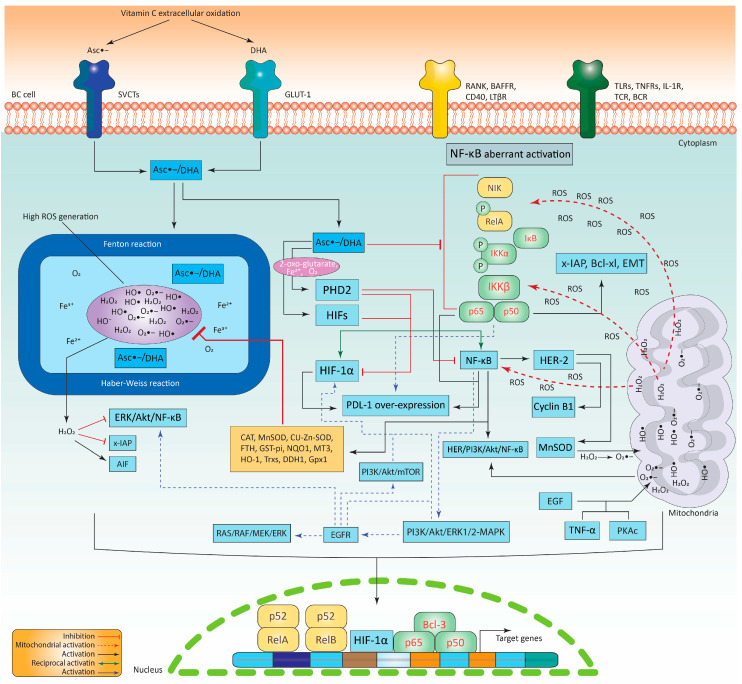
High-dose vitamin C targeting different pathways of the NF-κB and HIF-1α. The NF-κB can take advantage of ROS generated by mitochondria to activate various genes such as HIF-1α in favor of BC. Vitamin C, on the other hand, produces a high amount of ROS that can inhibit NF-κB and its various subunits directly or indirectly. Vitamin C activates enzymes that are able to inhibit NF-κB and HIF-1α as well as their target genes. BC cell, breast cancer cell; TLRs, Toll-like receptors; TNFRs, TNF-α receptors; TCR, T cell receptor; BCR, B cell receptor; LTβR, lymphotoxin β receptor; BAFFR, the BAFF receptor; RANK, receptor activator of nuclear factor-kappa beta; PI3K, phosphoinositide 3-kinase; ERK, extracellular signal-related kinase; HER2, human epidermal growth factor receptor 2; NF-κB, nuclear factor-kappa beta of activated B cell; NIK, NF-κB-inducing kinase; PD-L1, programmed cell death receptor 1 ligand; PHD-2, prolyl hydroxylase-2; FIH, factor-inhibiting HIF-1α; SVCTs, sodium-dependent vitamin C transporters; GLUT-1, glucose transporter 1; Fe^3+^, ferric ion; Fe^2+^, ferrous ion; ROS, reactive oxygen species; DHA, de-hydroascorbic acid; Asc^•−^, ascorbic intermediate radical; PKAc, protein kinase; CAT, catalase; MnSOD, manganese superoxide dismutase; Gpx1, glutathione peroxidase-1; CU-Zn-SOD, copper–zinc superoxide dismutase; FTH, ferritin heavy chain; Trx, thioredoxin; GST-pi, glutathione S-transferase Pi; NQO1, NADPH-dehydrogenase-quinone-1; MT3, metallothionein-3; EGF, epidermal growth factor; x-IAP, X-linked inhibitor of apoptosis; AIF, mitochondrial flavoprotein. Note: For more information, please see the text.

**Figure 4 biomedicines-11-01060-f004:**
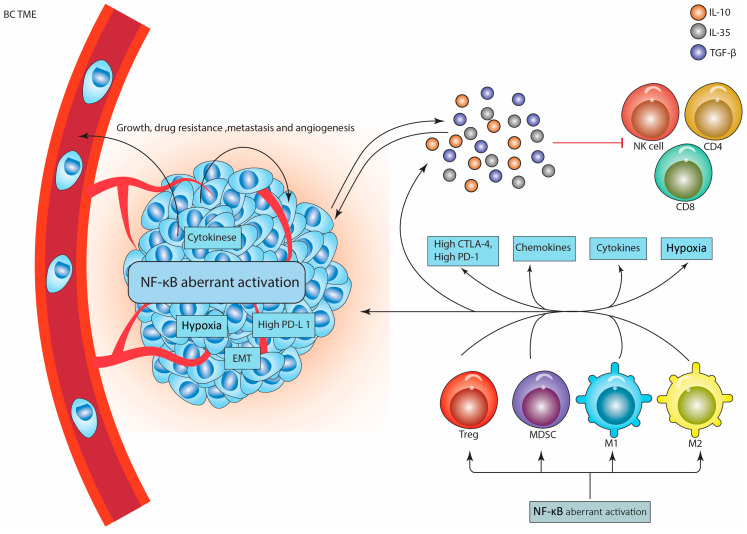
NF-κB aberrant expression recruits immunosuppressive cells in BC microenvironment. Following NF-κB activation in the immunosuppressive cells in addition to BC cells, together they contribute to promoting a highly immunosuppressive milieu, which will lead to BC development. The immunosuppressive cells produce various cytokines such as IL-10, IL-35, and TGF-β to suppress the anti-cancer immune response that is mediated by CD8, CD4, and NK cells. Aberrant activation of NF-κB in BC cells, in addition to the hypoxic conditions that enhance the activation of the HIF-1α, increases the expression of various materials that support BC growth, drug resistance, metastasis, and angiogenesis. In addition, the immunosuppressive cells and BC cells activate each other by secreting various cytokines, thus initiating a highly immunosuppressive microenvironment. NF-κB, nuclear factor-kappa beta of activated B cell; PD-L1, programmed cell death receptor-1 ligand; PD-1, programmed cell death receptor-1; CTLA-4, cytotoxic T lymphocyte antigen-4; TGF-β, transforming-growth factor-β; MDSCs, myeloid-derived suppressor cells; Tregs, regulatory T cells; M1, type 1 macrophage; M2, type 2 macrophage; BC TME, breast cancer tumor microenvironment; EMT; epithelial-to-mesenchymal transition; IL-10, interleukin 10; IL-35, interleukin 35. Note: For more information, please see the text.

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
