# Peer review of "The NF-κB Transcriptional Network Is a High-Dose Vitamin C-Targetable Vulnerability in Breast Cancer"

_biomedicines, 2023, doi:10.3390/biomedicines11041060_

Round 1

Reviewer 1 Report

Mussa et al. conducted a comprehensive review concerning the role of vitamin C in breast cancer.

Overall, I believe that this review is suitable for publication because it offers substantial amount of evidence in this regard, and offers admirable in-depth analysis between vitamin c and NfkB pathway. 

The biggest pitfall of the present study is lack of clinical studies. I think it would be appropriate that authors expand this part. Accordingly, in the last paragraph future directions should be discussed in light of clinical applicability of these findings. Specifically the authors should address the following questions: Is there a place for vitamin C in clinical management of patients with BC? What impeded us from doing so insofar? What should be done in the future in order to establish whether vitamin C is useful in this setting?

Minor point: define all abbreviations in the figure legends, it is hard to keep track without it.

3.2 subheading - there is a typo

The author should update the references as many of them are outdated

Author Response

Responses to Reviewer 1 Comments

We are appreciative to the reviewers for taking the time to assess our work and provide helpful feedback and legitimate criticism. We found them extremely useful in detecting flaws and ambiguities in our paper and carefully analyzed them. Accordingly, the manuscript has been revised. We believe that by doing so, the revised manuscript has been significantly enhanced. The following responses have been prepared to address all of the reviewers’ comments in a point-by-point fashion.

Comments and Suggestions for Authors

Mussa et al. conducted a comprehensive review concerning the role of vitamin C in breast cancer.

Overall, I believe that this review is suitable for publication because it offers substantial amount of evidence in this regard, and offers admirable in-depth analysis between vitamin c and NfkB pathway.

  1. The biggest pitfall of the present study is lack of clinical studies. I think it would be appropriate that authors expand this part. Accordingly, in the last paragraph future directions should be discussed in light of clinical applicability of these findings. Specifically, the authors should address the following questions: Is there a place for vitamin C in clinical management of patients with BC? What impeded us from doing so insofar? What should be done in the future in order to establish whether vitamin C is useful in this setting?

Response: Thank you for your insightful suggestions and comments. The point that was suggested by the reviewer is highly important, and we agree with it. However, the goal of this review is to shed light on the relevance and complexity of NF-κB in breast cancer formation, as well as to indicate how pro-oxidant chemicals like Vitamin C may assault these NF-κB networks, opening the way for clinical trials. Nevertheless, no clinical studies using high-dose vitamin C, especially in targeting NF-κB in breast cancer, have been done so far. Several pre-clinical and clinical investigations have confirmed the functions of high-dose vitamin C in cancer therapy, but the findings have been contentious, necessitating more research to determine the whole role of vitamin C in cancer treatment. To the best of our knowledge, this is the first thorough review that covers the effect of high-dose vitamin C in NF-κB network inhibition. We anticipate that this evaluation will pave the way for more pre-clinical and later clinical investigations that will aid in targeting the very complicated NF-κB networks in breast cancer.

  1. Minor point: define all abbreviations in the figure legends, it is hard to keep track without it.

Response: Thank you for your insightful suggestions and comments. The abbreviations through the entire manuscript were added accordingly to all the figures.

  1. 2 subheading - there is a typo

Response: Thank you for your insightful suggestions and comments. The typo is corrected accordingly. (Line: 127).

  1. The author should update the references as many of them are outdated

Response: Thank you for your insightful suggestions and comments. The authors agreed with the respected reviewer that some references are old. However, they are very critical in our discussion since they have fundamental information regarding NF-κB and Vitamin C.

Reviewer 2 Report

In this review, Nf-kB impact on breast cancer is studied.  Nf-Kb is usually studied in cytokines activity and its regulation by corticosteroids. Rarely its regulation is addressed to Ros.  This review inspires new perspectives about the role of Ros and antioxidants as vitamin C in the regulation of some genes expression, but it does not establish a definite role of Ros and vitamin C in the activity of Nf-Kb, as Ros can simultaneously activate or inhibit Nf-Kb. 

I would probably further discuss further future possible implications .

Author Response

Responses to Reviewer 2 Comments

We are appreciative to the reviewers for taking the time to assess our work and provide helpful feedback and legitimate criticism. We found them extremely useful in detecting flaws and ambiguities in our paper and carefully analyzed them. Accordingly, the manuscript has been revised. We believe that by doing so, the revised manuscript has been significantly enhanced. The following responses have been prepared to address all of the reviewers’ comments in a point-by-point fashion.

Comments and Suggestions for Authors

  1. In this review, NF-κB impact on breast cancer is studied.  NF-κB is usually studied in cytokines activity and its regulation by corticosteroids. Rarely its regulation is addressed to ROS.  This review inspires new perspectives about the role of ROS and antioxidants as vitamin C in the regulation of some gene’s expression, but it does not establish a definite role of ROS and vitamin C in the activity of NF-κB, as ROS can simultaneously activate or inhibit NF-κB. 

Response: Thank you for your insightful suggestions and comments. We began (Lines 500-536) by providing thorough information on how ROS activate NF-κB proteins. Then, in the next part, we explained in detail how large levels of ROS produced by high-dose vitamin C may efficiently inactivate NF-κB.

  1. I would probably further discuss further future possible implications.

Response: Thank you for your insightful suggestions and comments. A discussion statements have been added in conclusion and feature perspectives.

Reviewer 3 Report

-          Fig2 has not been shown.

-          According to the title the authors try to link the high dose of vitamin C with NF-κB activation in the treatment of breast cancer.

-          They first expose the NF-κB activation pathways in BC and second the pro-oxidant activity of vitamin C. But in line 472 they state: “Using different antioxidants including vitamin C effectively inhibited the activation of NF-κB [108, 109]”.  It should clarify what type of activity influences de the activation of NF-κB the pro-oxidant or antioxidant activity of vitamin C.

-          References 105 and 119 are the only ones that show the relationship between NF-κB and vitamin C. If the title, try to relate both, it could be not enough to propose as a potential vulnerability to treat BC.

Author Response

Responses to Reviewer 3 Comments

We are appreciative to the reviewers for taking the time to assess our work and provide helpful feedback and legitimate criticism. We found them extremely useful in detecting flaws and ambiguities in our paper and carefully analyzed them. Accordingly, the manuscript has been revised. We believe that by doing so, the revised manuscript has been significantly enhanced. The following responses have been prepared to address all of the reviewers’ comments in a point-by-point fashion.

Comments and Suggestions for Authors

  1. Fig2 has not been shown.

Response: Thank you for your insightful suggestions and comments. The authors apologies for this mistake, Figure 2 has been added accordingly.

  1. According to the title the authors try to link the high dose of vitamin C with NF-κB activation in the treatment of breast cancer. They first expose the NF-κB activation pathways in BC and second the pro-oxidant activity of vitamin C. But in line 472 they state: “Using different antioxidants including vitamin C effectively inhibited the activation of NF-κB [108, 109]”.  It should clarify what type of activity influences the activation of NF-κB the pro-oxidant or antioxidant activity of vitamin C.

Response: Thank you for your insightful suggestions and comments. Here NF-κB utilized TNF-α-mediated ROS activation, thus, vitamin C was used to inhibit ROS through its antioxidant activity, and the point was how low concentrations of ROS activate NF-κB. However, the pro-oxidant activity of vitamin C produces very high concentrations of ROS to inactivate NF-κB.

  1. References 105 and 119 are the only ones that show the relationship between NF-κB and vitamin C. If the title, try to relate both, it could be not enough to propose as a potential vulnerability to treat BC.

Response: Thank you for your insightful suggestions and comments. All the references in the review were chosen based on their relevance to the suggested topic. For example, 116, 117, and 121 are highly relevant to the current topic. The remining references are highly important to corelate the topic together, for instance, correlation between the NF-κB, PD-L1, and HIF-1α, and how they utilize ROS to induce breast cancer development.

Round 2

Reviewer 3 Report

Now it can be accepted

Author Response

Thank you for your nice coment.